# GAMMA: Graph Neural Network-Based Multi-Bottleneck Localization for Microservices Applications

## ABSTRACT

Microservices architecture is quickly replacing monolithic and multi-tier architectures as the implementation choice for large-scale web applications as it allows independent development, scalability, and maintenance. However, even with careful node scheduling and scaling, the microservices applications are still vulnerable to performance degradation due to unexpected (dependent or independent) events like anomalous node behavior, workload interference, or sudden spikes in requests or retries. These events can adversely affect the performance of one or more microservices (bottlenecks), degrading the overall application performance. To ensure a good customer experience and avoid revenue loss, it is crucial to detect and mitigate all bottlenecks swiftly.

This work introduces GAMMA, a novel, explainable graph learning model that integrates a mixture of experts to detect multiple bottlenecks. We evaluated GAMMA using a popular open-source benchmarking application deployed on Kubernetes under various practical bottleneck scenarios. Our experimental evaluation results show that GAMMA provides significantly better performance (46% higher $F_1$ score) than existing works that employ deep learning, machine learning, and statistical techniques, demonstrating its ability to detect multiple bottlenecks by learning complex interactions in a microservices architecture.

## 1 INTRODUCTION

*Microservice architecture (MSA)* is quickly becoming the choice of implementation for large-scale web applications owing to its modular nature [1, 6, 15, 20, 28, 29, 32, 46–48, 52]. Indeed, MSA is replacing monolithic and multi-tier application architectures as MSA designs applications as fine-grained, modular, and independent services called microservices, enabling independent development, scalability, and maintenance [15, 20, 28]. Even existing web and online gaming applications implemented using monolithic architecture are being transformed to MSA [1, 13, 17, 25, 31].

A critical problem for online web applications is performance management as it affects customer experience and revenue [7]. Among various aspects of performance management, detecting performance degradation and identifying the sources of performance degradation are crucial for providing a consistent user experience. We define *anomaly detection* as the process of detecting an application's performance degradation at the level of individual requests or over a time period, and *bottleneck localization* as the process of identifying which specific microservices are affecting the application's performance. Despite careful application design and proactive capacity planning, performance anomalies still happen due to unexpected surges in load or workload interference [20, 28, 48]. For that reason, bottleneck localization is a must. The microservices with degraded performance, i.e., *performance bottlenecks*, often arise due to resource saturation, resource contention, or microservice

application misconfiguration [15, 16, 38, 44], and do not necessarily lead to errors or faults, making them difficult to detect.

Anomaly detection and bottleneck localization in MSA applications are challenging for various reasons. Firstly, bottlenecks can manifest in *different ways*, impacting one or more microservices, even propagating across microservices over time and sustaining even after the source of anomaly is mitigated [15, 19, 43]. This exacerbates bottleneck localization, multiplies the engineering hours needed to mitigate with time, and delays the restoration of applications' performance. Secondly, the effect of performance anomalies (e.g., host interference) differs across microservices. For example, two microservices hosted on the same node that is experiencing CPU saturation will react differently in terms of degradation depending on how compute-bound the microservices are. This necessitates solutions that learn the *unique characteristics* of the microservices. Thirdly, complex interactions among the microservices can complicate bottleneck localization. The dynamicity that arises from asynchronous calls, caching, queues, feature additions, deprecations, and design changes can further complicate these interactions. As such, the solution must utilize and learn from the *interactions* of these microservices. Lastly, the absence of publicly available datasets with metrics, traces, and logs containing multiple bottlenecks from various sources has hindered the ability of researchers to evaluate their methods for multi-bottleneck detection and localization [24].

The key challenge, and the focus of this paper, is the presence of **multiple bottlenecks** in MSA applications. Existing works, including those in recent editions of The Web Conference, have primarily focused on single bottlenecks and have ignored the practical case of multiple bottlenecks [29, 41, 50]. There are different ways in which multiple bottlenecks can arise in practice.

- Multiple, *independent* bottlenecks arise in one or more microservices. For example, a microservice responsible for logins could be bottlenecked due to a sudden spike in user logins, while simultaneously, another microservice could be bottlenecked due to resource contention at its host node. In such cases, all bottlenecks must be (independently) detected and mitigated.
- Multiple, *dependent* bottlenecks arise in one or more microservices, due to the same underlying problem. For example, if the underlying VM that hosts multiple microservices is under resource contention from different colocated VMs, then all microservices on this VM can experience performance degradation.
- Multiple, *cascading* bottlenecks appear in sequence in multiple microservices. For example, a database microservice that is experiencing workload interference can result in request queues building up in dependent microservices, causing their performance to degrade as well. Undetected, these bottlenecks can cascade to interacting microservices, increasing the number of bottlenecks over time. In such cases, it is important to first detect

all bottlenecks, and then alleviate them to quickly revive the application performance.

Prior works in this space have mainly focused on anomaly detection [26, 50, 51] or providing solutions for single bottleneck localization that cannot be easily adapted for multiple bottleneck localization [14, 16, 18, 24, 29]. Even solutions that are capable (with some effort) of detecting multiple bottlenecks are not evaluated on traces or datasets with multiple bottlenecks [38, 41]. Moreover, the latter works [38, 41] fail to detect multiple bottlenecks effectively (Section 4) as they do not utilize the distributed traces to learn the complex interactions among microservices. To the best of our knowledge, *ours is the first work that evaluates multi-bottleneck anomaly detection and bottleneck localization for MSA.*

This work introduces and evaluates GAMMA, a novel model to **detect anomalies and multiple bottlenecks** in web applications implemented using the microservices architecture. Specifically, GAMMA uses (a) an attention-based graph convolution network to *learn the complex interactions* between microservices, (b) a holistic multi-source end-to-end joint training framework to detect the presence of bottlenecks in an *explainable* manner, and (c) a mixture of experts to account for possibly *multiple bottlenecks* across microservices.

In designing and evaluating GAMMA, this work makes the following key contributions:

(1) We present the design of GAMMA, a holistic multi-source end-to-end joint training framework that learns complex interactions between microservices using an attention-based graph convolution trained over distributed traces of observable metrics (e.g., CPU and memory utilization), which are readily available in production systems [30]. Further, it uses a mixture of experts to learn the unique characteristics of microservices and account for possibly multiple bottlenecks across microservices.

(2) To evaluate GAMMA, we generate a dataset consisting of around 40 million request traces; we commit to open-sourcing the dataset to aid research in this area. The dataset, created using a popular open-source benchmarking application [15], consists of multiple bottlenecks from various sources while serving workloads of different intensities.

(3) We evaluate GAMMA against existing techniques on the above bottleneck dataset; we also extend a seminal prior work [16] created for localizing single bottlenecks to localize multiple bottlenecks.

(4) We perform a detailed ablation study to understand and explain the impact of telemetry on evaluation results.

Our experimental evaluation results show that GAMMA provides an $F_1$ score of up to 0.92 and 0.89 for anomaly detection and bottleneck localization, respectively. GAMMA significantly exceeds the performance of prior works (3–4× improvement for anomaly detection and 46% improvement for bottleneck localization) based on deep learning, machine learning, and statistical techniques, demonstrating its ability to detect multiple bottlenecks by learning complex interactions in microservices architecture.

Our analysis reveals that the performance gap between GAMMA and other baselines increases with the increasing complexity of the evaluation scenario. While existing works perform reasonably well when there is a single source of anomaly, their performance drops when evaluated in scenarios consisting of multiple sources of anomaly, unlike GAMMA. Further, while existing works can perform better if they are separately trained on each source of anomaly, GAMMA provides consistently better performance despite not being trained separately on individual anomaly sources, making GAMMA easier to deploy in practice. Finally, we show that GAMMA can provide explainability with its bottleneck localization, thereby aiding the bottleneck mitigation task.

## 2 RELATED WORK

Related prior works can be broadly categorized into (a) anomaly detection works, and (b) bottleneck localization (or root cause analysis) works. Since there are numerous prior works in these general areas, we limit our discussion below to closely related works and refer readers to relevant surveys for further detail [42].

### 2.1 Anomaly Detection

DeepTraLog [51] uses a unified graph embedded with log events, called trace event graphs, to represent the complex interaction among microservices. It finds anomaly scores for each trace or request by training a gated graph neural network-based deep support vector data description model on the trace event graphs. Trace-VAE [50] is an unsupervised anomaly detection model that uses a novel dual-variable graph variational autoencoder with Negative Log-Likelihood (NLL) as the anomaly score. TraceAnomaly [26] is an unsupervised anomaly detection system that uses novel trace representation and deep Bayesian networks with posterior flow. The model is trained offline periodically to learn normal patterns in traces and then classifies traces as anomalous when they deviate from these learned patterns.

### 2.2 Bottleneck Localization

Groot [47] is Ebay's graph-based framework for bottleneck localization in MSA applications. Groot constructs a causality graph with events that include anomalies in metrics, abnormal log statements, etc., as the nodes and causal links between these nodes are based on domain knowledge. However, Groot requires domain knowledge for creating links between nodes and additionally requires continuous human involvement to track changes to the causal links between nodes. CRISP [52] is Uber's tool for critical path analysis over traces from MSA applications which can be used for anomaly detection and bottleneck localization. The critical paths in MSA, however, are dynamic [38], requiring constant recomputation of critical paths. Murphy [18] is an automated performance diagnosis system that detects bottlenecks in complex enterprise environments by monitoring data to define associations between entities in an MSA application. However, Murphy uses a linear model that cannot capture the complexities in production microservices [18, 19].

FIRM [38] proposes a Support Vector Machine (SVM) model for detecting bottlenecks on the critical path in the call graph. The SVM model is trained on hand-crafted features that capture the per-critical-path and per-microservice performance variability. FIRM only considers latency as a feature and also ignores the structural information in the call graphs of the MSA application, limiting its ability to detect multiple bottlenecks (as we show in Section 4.4).

Seer [16] is an online performance debugging system that leverages deep learning to detect and mitigate bottlenecks in MSA. Seer uses a hybrid network of Convolutional Neural Networks (CNN) and Long Short-Term Memory (LSTM) networks to learn spatial and temporal patterns that lead to bottlenecks. However, analysis of Alibaba's production systems suggests that CNN-based approaches fail to characterize complex graph dynamics and do not apply to real-world applications; instead, the authors suggest using GNNs [28]. Our evaluation of Seer on a dataset consisting of multiple bottlenecks further substantiates this claim (Section 4.4).

$\epsilon$-diagnosis [41] uses a threshold technique to detect anomalies and distance correlation [45] to compare metrics of anomalous traces and normal traces for localizing bottlenecks. The localization algorithm runs on each microservice without utilizing any structural information available through distributed tracing. As we show in Section 4.4, this and other drawbacks significantly impact the performance of $\epsilon$-diagnosis in the case of multiple bottlenecks.

AutoMAP [29] relies on a heuristic algorithm using forward, self, and backward random walks on a graph representing the interaction between services to localize bottlenecks. Since it is a heuristic, AutoMAP may not be accurate and can suffer for large call graph sizes [4, 52]. B-MEG [43] is a two-staged graph-learning-based classifier that does anomaly detection and bottleneck localization in the first and second stages, respectively. However, B-MEG is only designed to detect single bottlenecks. Eadro [24] is a framework that uses traces, logs, and metrics along with multiple models to learn representations, which in turn are used to detect anomalies and localize bottlenecks jointly. The framework, owing to the series of models it uses, makes it difficult to interpret the results. MicroCU [21] is a framework that uses API logs and Granger causality to detect bottlenecks. Ablation studies on the importance of telemetry, traces, and logs in detecting bottlenecks reveals that logs provide the least information to detect bottlenecks [24]. Sage [14] uses a Causal Bayesian Network (CBN) to capture the dependencies between microservices. However, the assumption in Sage that the latency of non-leaf nodes in the call graph is determined by the wait time of its child nodes might not always hold (e.g., when a non-leaf child node is a message queue [28]). Moreover, Sage [14] can only work on call graph DAGs (no cycles), but call graphs in production systems have cycles [18, 28].

In summary, prior works provide solutions or evaluate their solutions **only for single bottlenecks** [14, 16, 24, 29, 38, 41, 43, 47] and do not fully utilize the rich telemetry and distributed tracing that is part of the MSA [21, 30, 38, 41]. Our work, described next, addresses this important gap by using a graph learning module to understand the complex interaction among microservices and a mixture of experts model to detect multiple bottlenecks effectively.

## 3 DESIGN OF GAMMA

Traditional bottleneck localization techniques (deep learning or heuristic-based) often operate in a linear or isolated manner, failing to capture the *dependencies and interactions* inherent in MSA [34]. Consider Figure 1, which shows a small subgraph of the entire social network call graph from DeathStarBench suite [15]. A simultaneous failure in Machines 3 and 4 will impact the corresponding *on-chain* RPC calls but will not affect the *off-the-chain* ones.

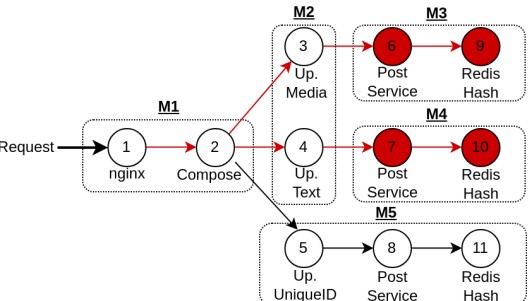

**Figure 1: A simultaneous failure in machines M3 and M4 will affect the RPC calls in the invocation chain while other RPC calls in the call graph are not affected.**

Graph Neural Networks (GNNs) are ideally suited to model such intricacies in graphical data and to capture dependencies between nodes [16, 43]. GNNs are designed to naturally assimilate and process information from nodes and their respective neighborhoods in a graph. GNNs can also handle the complexities of enterprise environments, especially cycles in the call graphs [49].

*The key idea behind GAMMA is to understand patterns in call graphs using inputs from multiple system metrics and the graph dependency structure*; this information can help identify the interconnections among microservices and guide system diagnosis. Figure 2 shows the architectural overview of GAMMA, which is broadly divided into 4 stages: Multi-Source Temporal Embeddings Learning, Graph-Representation Learning, Anomaly Classifier, and Bottleneck Localizer.

### 3.1 Multi-Source Temporal Embeddings Learning

Capturing temporal patterns helps to reveal the dynamic nature of system performance, highlighting fluctuations and evolving trends over time. Since bottlenecks may be induced due to episodic anomalies in the system, analyzing temporal chunks allows us to capture *correlations and sequences* across requests while also providing macroscopic trends in the system for the anomalous episodes [10, 51]. Multi-input temporal embeddings encapsulate the spatio-temporal behavior of a system, providing a comprehensive view of spatial relations and time-evolving patterns within a given window. Consider a call-graph with $\eta$ microservices. For the microservices, we organize the system metrics into an $\eta$-dimensional time-series $\mathbb{M}^\eta$. The system metrics (e.g., RPC latency, CPU usage) act as our model features. We split the entire feature tensor $\mathbb{M}^\eta$ into windows of length $\tau$, thus giving us window inputs of size $\mathbb{M}^{\eta \times \tau}$. The parameter $\tau$ is trainable and is decided based on validation metrics during training. Analyzing windows as opposed to individual traces allows us to aggregate the temporal dynamics of the system.

The input tensor is processed using a Multivariate Temporal Convolution Attention Network, which is designed to recognize patterns over time. This network employs Dilated Causal Convolution (DCC), a method that efficiently captures relationships within and between features over time. DCC is highly scalable, and it has proven to be superior to traditional methods, like CNNs and LSTMs,

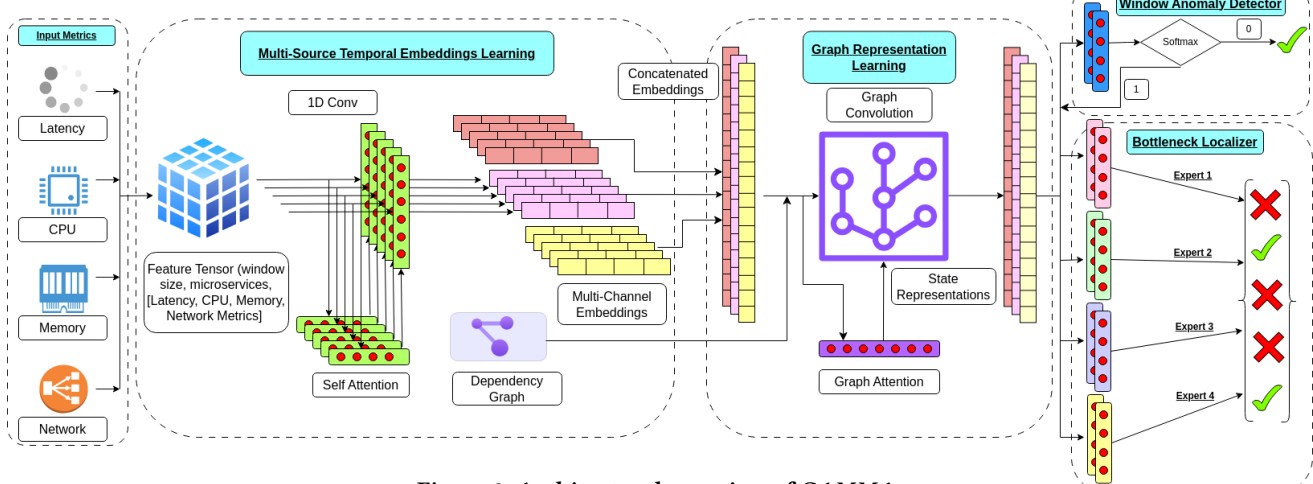

**Figure 2: Architectural overview of GAMMA.**

especially when predicting future events based on past data. The dilated causal convolution for a feature vector $\mathbb{M}^{\eta x \tau}$ is

$$\mathbb{O}(\eta, k, q) = \sum_{\eta} \sum_{\tau + \delta \cdot s = q} \mathbb{M}(\eta, \tau) \times Weight(k, s), \quad (1)$$

where $\mathbb{M}(\eta, \tau)$ is the input tensor, $\mathbb{O}(\eta, k, q)$ are the multi-channel output embeddings, $k \times q$ represent the Convolution filters, $\delta$ is the expansion factor, and $Weight(k, s)$ is the filter size for $\eta$ output channels. Self-attention is then applied on the input embedding tensor $\mathbb{M}^{\eta x \tau}$, as:

$$Attn(\mathbb{M}) = softmax\left(\frac{W_q\mathbb{M} \cdot (W_k\mathbb{O})^T}{\sqrt{d}} W_v\mathbb{M}\right), \quad (2)$$

where $W_q$, $W_k$ and, $W_v$ are trainable hyperparameters and $d$ is an empirical scaling factor. This phase outputs multi-channel embeddings with latent representation $\mathbb{O}^{\eta \times k \times q}$.

## 3.2 Graph Representation Learning

In this stage, our goal is to understand the end-to-end status of the MSA application and provide a detailed overview of the entire system including the dependent interactions between the microservices themselves. This requires three key actions: (1) merging the multi-channel embeddings generated in the previous stage to get concatenated embeddings for each microservice; (2) incorporating the microservice call-graph and the concatenated embeddings to generate the dependency graph and microservice-level status representations (node representations) for the application; and (3) modeling this dependency graph. We begin by creating a directed graph from the call-graph that illustrates how microservices are interconnected. Next, we integrate the output embedding sourced from earlier stages into unified node representations, showcasing the status at the microservice-level. Information within this graph is then channeled through a GNN, enabling the understanding of neighboring interconnections and interactions.

*3.2.1 Generating the Dependency Graph.* The process of extracting a call graph from microservices traces can be systematically understood by visualizing microservices as nodes and their invocations

as directed edges. A dependency graph $G = \{V, E\}$ can be derived from traces, where $V$ denotes the set of nodes with $|V| = M$, with $M$ being the total number of distinct microservices. $E$ represents the set of edges; an edge $e_{a,b} = (v_a, v_b) \in E$ indicates a directed relationship from node $v_a$ to node $v_b$, implying that the microservice associated with node $v_b$ has made an invocation to that associated with node $v_a$ at least once in recorded history.

Since it is essential for us to calculate temporal representations of microservices that capture both inter- and intra-feature correlations for our inputs, we *concatenate* our embeddings at an intermediate stage before we generate our dependency graph [22, 24]. Studies in cross-modal learning [23, 27, 33] hint that intermediate fusion tends to be more effective for processing temporal representations. Initially, we concatenate $([\cdot \| \cdot])$ the representations of each microservice acquired from the prior phase, ensuring comprehensive data retention. The resulting tensor is then projected on a lower dimensional subspace by passing it through a fully-connected layer and subsequently passed through a Gated Linear Unit to fuse the representations while controlling for vanishing gradients and increasing resiliency to gradient forgetting [12]. The microservice levels concatenated embeddings $\mathbb{O}^{\eta \times \epsilon}$ serve as node embeddings for the GNN with each node $\eta_n$ having the embedding vector $\mathbb{O}^{\epsilon}_{\eta_n}$.

*3.2.2 Graph-Attention Network.* We employ the Graph Attention Network (GAT) [9], a specialized GNN variant that offers several advantages in the context of microservices [24]. Unlike traditional GNNs, GAT is capable of learning node and edge representations while dynamically assigning importance weights to neighboring nodes. This attention mechanism ensures that the network can *focus on more influential or anomalous microservices*, potentially acting as communication hubs or displaying abnormal behavior patterns. The local representation $\mathbb{O}^{\epsilon}_{\eta_n}$ encapsulates the feature set for individual nodes. The model digests this information and learns a holistic representation of the entire graph. Dynamic edge weights, integral to the attention mechanism, are formulated as per Equation (3),

ensuring an understanding of microservice interactions.

$$\omega_{a,b} = \frac{exp(LeakyReLU(v^T[W\mathbb{O}_a^\epsilon||W\mathbb{O}_b^\epsilon]))}{\sum_{k \in \mathbb{N}_a} exp(LeakyReLU(v^T[W\mathbb{O}_a^\epsilon||W\mathbb{O}_k^\epsilon]))}, \qquad (3)$$

where $\omega_{a,b}$ is the computed weight of edge $\vec{e}_{a,b}$, $\mathbb{N}_a$ is the set of neighbor nodes for node $a$; $\mathbb{O}_a^\epsilon$ is the intermediate node representation of node $a$; $W \in \mathbb{R}^{E^G \times E}$ and $v \in \mathbb{R}^{E^G}$ are trainable parameters. $E^G$ is the shape of the output representation. The impact of all the neighboring nodes $b$ on node $a$ is calculated as follows:

$$\hat{O}_a^\epsilon = ReLU \sum_{b \in \mathbb{N}_a} \omega_{a,b} W\mathbb{O}_b^\epsilon \qquad (4)$$

Global Attention Pooling [8] is then performed on the node representations to generate dependency-aware embeddings $\mathbb{O}^\zeta$.

## 3.3 Detection and Localization

In the final phase, GAMMA performs two functions: it predicts if a given observation window indicates an anomaly (anomaly detection), and if so, it discerns which microservices are the root cause (bottleneck localization). Contrary to traditional approaches [16, 38] which treat anomaly detection and bottleneck localization as separate functionalities, GAMMA adopts a holistic approach to leverage the knowledge of the inter-related functionalities.

Leveraging the earlier acquired representation $\mathbb{O}^\zeta$, an initial detector performs a binary assessment to ascertain the presence of any anomalies. If the outcome is negative, GAMMA directly presents the results. However, if an anomaly is detected, a subsequent localizer arranges the microservices in order of their likelihood to be the origin of the issue. This two-step mechanism, comprising the detector and the localizer, employs multiple experts comprising of connected neural networks followed by a binary classifier. Each microservice in the call-graph has a dedicated expert assigned to predict if the microservice is bottlenecked or not. Both these components, the detector and localizer, are trained in tandem with a shared goal. The model's primary focus is to curtail the total binary cross-entropy loss of the detector ($\lambda_d$) and localizer ($\lambda_l$). The joint loss function is given as:

$$\lambda_{total} = \alpha \cdot \lambda_l + \sum_{k \in \eta} \frac{(1-\alpha)}{\eta} \cdot \lambda_k, \qquad (5)$$

where $\lambda_l = \sum_{k \in \eta} \lambda_k$; and $\alpha$ is a hyperparameter to tune the contribution of $\lambda_l$ and $\lambda_d$ towards the total loss. Should an anomaly be detected, GAMMA outputs a binary vector of 0s and 1s which predicts the bottleneck and non-bottlenecked microservices.

## 4 EVALUATION

We now present our experimental evaluation results for GAMMA under various bottleneck scenarios. We also compare GAMMA's performance with that of recent works on bottleneck localization.

## 4.1 Experimental Setup

We evaluate GAMMA on a cluster of 17 VMs (4 vCPUs, 8GB memory) managed by Kubernetes. The VMs are synchronized via NTP for accurate measurements. The metrics (CPU, memory, network) are collected via Prometheus [39], while Jaeger [3] collects distributed traces. To generate a variety of bottlenecks, we use a CPU load generator [2] and *stress-ng* tool to generate interferences on one or more host VMs. This generates multi-bottlenecks of varying intensities and duration that may overlap in time.

We use the popular social networking benchmark from Death-StarBench [15] that consists of 28 microservices implementing several features of real-world social networking applications. The constituent microservices are Nginx, Memcached, MongoDB, Redis, as well as microservices that implement the logic of the application. The workload consists of *Compose* requests that create a post, *User* requests that read the timeline of other users, and *Home* requests that read the user's own timeline. We use wrk2 [5] to generate workloads of different intensities. We benchmark the application to find the peak load (800 requests per second, or RPS) beyond which it is unstable. We use different intensities in the range of 100–800 RPS. We deploy monitoring services like Prometheus and Jaeger on a separate VM to avoid unintended interference.

## 4.2 Dataset Creation

A key contribution of this work is constructing a dataset for research on anomaly detection and multi-bottleneck localization. Prior works have noted that existing public traces [37] on anomaly detection and bottleneck localization only contain single, severe bottlenecks that are not representative of real-world scenarios [43]. When such a bottleneck is introduced, the resulting latency increases by an order of magnitude (100×), making it trivial to detect that singe bottleneck using a simple grid search or threshold-based approaches.

To create a more realistic dataset that includes traces with *multiple bottlenecks* at *different intensities*, we carefully benchmarked the social networking application under different interference intensities and duration of interference. We chose intensities and duration values that degrade the application performance but do not cause any faults or errors that can be trivially detected. We induced interference on different VMs at different times and also simultaneously. A single VM could be induced with different types of interference (e.g., CPU and memory), resulting in the hosted microservices experiencing a mixture of interference patterns. The resulting dataset consists of around *40 million request traces* along with corresponding time series of CPU, memory, I/O, and network metrics. The dataset also includes application, VM, and Kubernetes logs. We commit to open-sourcing the raw and processed datasets, along with our GAMMA implementation, to further research in this important area.

## 4.3 Metrics and Baselines

For evaluation of anomaly detection and bottleneck localization, we use the following performance metrics:

- *Recall* is the ratio of true positive predictions to the total number of positive data points. It measures how many of the positive data points were classified as positive by the model. A high recall is essential for MSA-based web application deployments as it is important to detect *all* anomalies and bottlenecks.
- *Precision* is the ratio of true positive predictions to the total positive predictions. It measures how many of the data points that were classified positive by the model are actually positive.

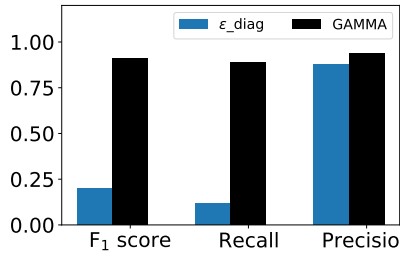

(a) Compose request type.

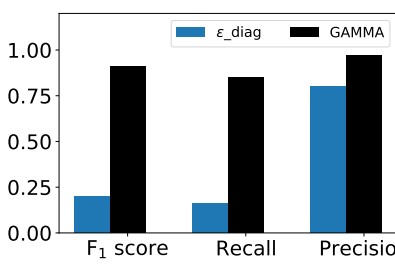

(b) User request type.

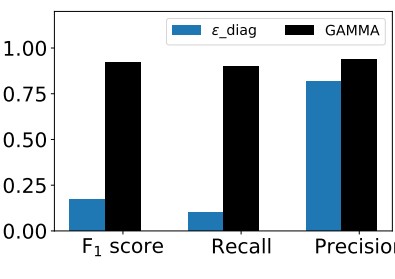

(c) Home request type.

Figure 3: $F_1$ score, Recall, and Precision for anomaly detection over the entire dataset.

A high precision is desirable as it implies fewer engineer hours wasted investigating false positives.

- $F_1$ *score* is the harmonic mean of precision and recall. It is a metric that balances the trade-off between precision and recall.

We experimentally compare the performance of GAMMA with the following state-of-the-art baselines from recent works:

(1) *FIRM* is a framework that uses SVM and hand-crafted features to localize bottlenecks on the critical path. We use Scikit-learn library [36] to implement FIRM's SVM model.

(2) *$\epsilon$-diagnosis* performs both anomaly detection and bottleneck localization. It uses a simple threshold scheme for anomaly detection and distance correlation for bottleneck localization. We use the dcor [40] library to implement the $\epsilon$-diagnosis' localization module.

(3) *Seer* is an online bottleneck localization framework that uses CNN and LSTM to learn spatial and temporal features, respectively, to recognize patterns that lead to anomalies. We implement Seer using Pytorch [35].

(4) *Seer\** is our modified version of Seer for multi-bottleneck localization which works by adapting softmax to individual binary classification for each microservice in the call-graph and replacing cross-entropy loss with hinge-loss [11].

To evaluate $\epsilon$-diagnosis and FIRM on multi-bottleneck data, we run these baselines on all the microservices serially. Since the original Seer model cannot be directly applied for multi-bottleneck localization, we evaluate how well it localizes the most dominant bottlenecked microservice. We tune the hyperparameters of all baselines and present the best results in our evaluation.

## 4.4 Results

*4.4.1 Aggregate results for anomaly detection.* We start by evaluating GAMMA and the baselines using our entire dataset (with all resource bottleneck traces). Figure 3 shows the $F_1$ score, Recall, and Precision for anomaly detection using the entire trace dataset for GAMMA and $\epsilon$-diagnosis. Note that $\epsilon$-diagnosis is the only baseline among those considered that does anomaly detection. Starting with Figure 3a, which shows the results when analyzing Compose traces, we see that GAMMA provides significantly better results than $\epsilon$-diagnosis. The $F_1$ score, Recall, and Precision values for GAMMA are 0.91, 0.89, and 0.94, respectively.

By contrast, the corresponding values for $\epsilon$-diagnosis are lower by 78%, 87%, and 6%, respectively. We do observe that $\epsilon$-diagnosis

achieves reasonable Precision because of the low confidence threshold that its localizer uses, which ensures the quality of predictions. Recall, from Section 2, that $\epsilon$-diagnosis does not leverage any structural information about the application, thus losing out on important information. Further, $\epsilon$-diagnosis uses a static threshold to detect anomalies. While this threshold might work well for scenarios where only a single, severe performance bottleneck exists, this static threshold does not adapt to the more realistic case of multiple, different bottlenecks. In fact, when we evaluated $\epsilon$-diagnosis for the simpler, pathological dataset where a single bottleneck exists that causes performance to degrade significantly [38], $\epsilon$-diagnosis resulted in near-perfect $F_1$ scores. This underscores the difficulty in anomaly detection when multiple bottlenecks exist.

Results are similar for User (Figure 3b) and Home (Figure 3c) requests, with GAMMA significantly outperforming $\epsilon$-diagnosis and achieving high performance values. Specifically, in Figure 3b, GAMMA's $F_1$ score (0.91) is 355% higher than that of $\epsilon$-diagnosis (0.20). Likewise, in Figure 3c, GAMMA's $F_1$ score (0.92) is 441% higher than that of $\epsilon$-diagnosis (0.17). We note that User and Home requests have smaller call graphs than Compose. Additionally, Compose has asynchronous calls, caches, queues, and other complexities, that are inherent in MSA applications, making Compose a popular choice for analysis in prior works [24, 38]. While we experimented with all request types, due to lack of space, we will primarily focus on the complex Compose request type in our results.

*4.4.2 Aggregate results for bottleneck localization.* Figure 4 shows our results for the more challenging bottleneck localization task using the entire trace dataset for GAMMA and all baselines. *Across all request types, GAMMA outperforms all other baselines for all performance metrics.* In particular, GAMMA achieves a high $F_1$ score of 0.83–0.87 across Figures 4a–4c. Further, GAMMA also achieves a high Recall of 0.77–0.84 and a high Precision of 0.90–0.92 across all subfigures.

Starting with Figure 4a, we see that GAMMA outperforms all other baselines under all metrics. GAMMA achieves an $F_1$ score, Recall, and Precision of 0.83, 0.77, and 0.91, respectively. $\epsilon$-diagnosis again performs poorly, with an $F_1$ score of only 0.1; this is due to the weaknesses of $\epsilon$-diagnosis identified above which limit its accuracy for the multi-bottleneck scenario.

FIRM performs better than $\epsilon$-diagnosis, but still only achieves an $F_1$ score of 0.57 compared to the 0.83 (46% higher) obtained by GAMMA. This is likely because FIRM does not leverage the structural information in the call graphs of the MSA application

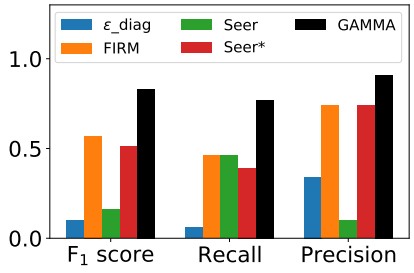

(a) Compose request type.

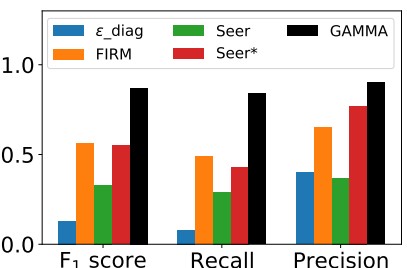

(b) User request type.

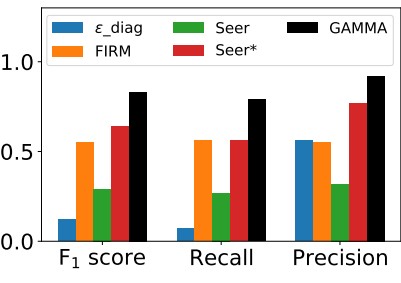

(c) Home request type.

**Figure 4: $F_1$ score, Recall, and Precision for bottleneck localization over the entire dataset.**

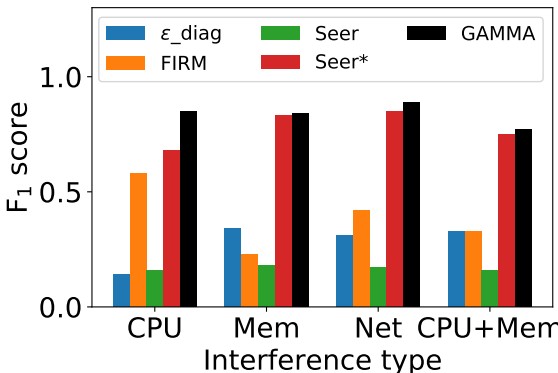

**Figure 5: Bottleneck localization results by interference type.**

or resource metric timelines, unlike GAMMA. We note that FIRM can perform quite well if we only consider single bottleneck traces, again highlighting the challenge of dealing with multiple bottlenecks. When we evaluated FIRM for the simpler, pathological dataset where a single, severe bottleneck exists [38], FIRM resulted in a much higher $F_1$ score of 0.83 with a Recall and Precision of about 0.7 and 0.9, respectively.

Seer also performs poorly, with an $F_1$ score of only 0.16. This is to be expected, however, as the unmodified Seer only focuses on localizing one bottleneck. Since real-world request traces may contain multiple bottlenecks (e.g., our traces contain as many as 11 bottlenecks each), Seer's performance is limited. To account for this shortcoming, we extended Seer to Seer* by replacing softmax in the prediction layer with individual binary classification for every microservice in the call graph, and then replaced cross-entropy loss with hinge-loss [11], as discussed in Section 4.3. With this extension, Seer* performs better, with an $F_1$ score of 0.51. However, this is still significantly below GAMMA's $F_1$ score of 0.83. We believe this is because while Seer* does leverage multiple neural network models, it does not make use of GNNs, which are ideally suited to MSA application call graphs [28]. As we show later in Table 3, the GNN component of GAMMA is crucial for good performance.

The results for User (Figure 4b) and Home (Figure 4c) request types are qualitatively similar to that of Compose in Figure 4a, with GAMMA outperforming all other baselines for all metrics.

*4.4.3 Results per bottleneck source.* We now evaluate GAMMA and the baselines by separately training and testing over traces that contain bottlenecks from a specific source (CPU, Memory, Network, CPU+Memory). This will allow us to assess the performance under specific bottleneck types. Figure 5 shows the $F_1$ score for GAMMA and all baselines for bottleneck localization (under Compose request type) separated by the interference type that creates the bottleneck. (GAMMA continues to be significantly better than $\epsilon$-diagnosis for anomaly detection so we omit those results.)

Across all subfigures, we see that GAMMA is always superior to the other baselines with an $F_1$ score of at least 0.77 and as much as 0.89 (for Network interference type). However, the performance of each baseline does differ across the subfigures. For example, FIRM performs much better when the bottlenecks are caused by CPU interference as opposed to other interference types. Seer* has the opposite behavior, with performance being close to that of GAMMA for non-CPU interference types, but worse for CPU interference. This suggests that specific baselines may perform better if they are separately trained for each source of bottleneck. However, this is tedious in practice. By contrast, GAMMA shows consistently good performance whether it is trained on each interference type (Figure 5) or more efficiently trained once on all interference types (Figure 4).

*4.4.4 Overhead analysis.* To compare the overhead of GAMMA and the baselines, we computed the average inference time across all traces for the combined tasks of anomaly detection and bottleneck localization, as applicable. Table 1 shows the overhead time in seconds for processing each window; for Seer*, which operates at the granularity of traces, we converted the times to per window by normalizing by the average number of traces in a window.

| GAMMA | $\epsilon$-diagnosis | FIRM | Seer* |
|---|---|---|---|
| $3.87 \times 10^{-5}$ | $2.75 \times 10^{-3}$ | $1.30 \times 10^{-6}$ | $5.78 \times 10^{-6}$ |

**Table 1: Average inference time (seconds) per window.**

We see that GAMMA has a much lower overhead than $\epsilon$-diagnosis, but is slower than FIRM and Seer*. Given the design of GAMMA, and its superior performance compared to FIRM and Seer*, we consider the larger inference time as a trade-off between performance and overhead. Regardless, we note that the overhead for GAMMA per 1s window is only about $38.7\mu s$, representing a 0.004% overhead for each second of window length.

*4.4.5 Explainability of GAMMA.* A key advantage of GAMMA is its ability to not just localize bottlenecks, but also aid in identifying the source of the bottlenecks. Utilizing multi-source data-based approaches, such as GAMMA, offers a significant advantage over other approaches that only rely on latency traces to identify and localize bottlenecks. System metrics, such as CPU usage and network congestion, can offer crucial insights, such as trends, threshold breaches, and correlations, in detecting bottlenecks for large MSA applications. GAMMA effectively integrates multiple system metrics with the microservice-dependency graph to understand cross-modal and temporal patterns for system interactions. To highlight this ability, we run GAMMA with different subsets of features. The intuition is that the feature whose omission causes a significant drop in performance is likely the source of the bottleneck.

| Feature Omitted | CPU interference | Memory interference | CPU+Memory interference |
|---|---|---|---|
| None | 0.851 | 0.844 | 0.771 |
| CPU | 0.693 | 0.805 | 0.714 |
| Memory | 0.789 | 0.573 | 0.600 |
| Network | 0.919 | 0.838 | 0.764 |
| CPU & Mem | 0.646 | 0.459 | 0.438 |

Table 2: Illustrating GAMMA's explainability by evaluating $F_1$ score when specific features are omitted from GAMMA.

The rows in Table 2 show the $F_1$ score when GAMMA is run with a specific subset of features for bottleneck localization. We consider three different bottleneck source scenarios, one per column: bottlenecks caused by (a) only CPU interference, (b) only Memory interference, and (c) CPU and Memory interference.

Starting with the only CPU interference scenario in the first column, we see that GAMMA's $F_1$ score drops from 0.851 to 0.693 when the CPU feature is omitted, but only drops to 0.789 when the Memory feature is omitted. When omitting the Network feature, the $F_1$ score actually increases to 0.919, suggesting that Network feature data may be hurting performance in this case. Overall, the results show that the CPU feature has a larger impact, suggesting that the source of bottleneck is CPU saturation. We see a similar result in the second column with the omission of the Memory feature creating a larger drop in $F_1$ score (0.844 to 0.573) compared to the omission of the CPU feature (0.844 to 0.805) or the Network feature (0.844 to 0.838).

In the final column, we see that omitting the CPU feature or omitting the Memory feature causes a reasonable drop in $F_1$ score, whereas omitting the Network feature only causes a small drop. However, dropping both features causes a much higher drop from 0.771 to 0.438. While omitting both CPU and Memory features is expected to cause a larger drop than omitting a single feature, the resulting drop for the final column is significant, suggesting that both CPU and Memory may have contributed to the bottleneck. Specifically, the additional drop in performance when omitting both features versus when only omitting the dominant feature is 7% and 20% for columns 1 and 2, respectively. By contrast, the drop is 27% for the last column.

*4.4.6 Ablation study for GAMMA.* The GAMMA core model involves a few stages, as discussed in Section 3. Two specific stages of

interest are the Graph Attention Network (which combines Graph Convolution with an attention mechanism allowing nodes to aggregate information from their most insightful neighbors) and Self Attention in context of causal convolution (which allows a temporal sequence to weigh the importance of its own past values). The former provides the ability to capture the interactions between microservices by leveraging the dependency structure. The latter ensures that the 1D-convolution operation, which is inherently local, is guided by a global understanding of the entire temporal sequence, ensuring a context-aware feature extraction.

| Stage Omitted | $F_1$ score | Recall | Precision |
|---|---|---|---|
| None | 0.830 | 0.87 | 0.83 |
| Graph Attention | 0.669 | 0.69 | 0.65 |
| Self Attention | 0.695 | 0.71 | 0.68 |

Table 3: Ablation study to highlight the importance of specific stages of GAMMA.

To validate our design choices, we performed an ablation study by replacing the two specific stages of GAMMA with alternative ones. Table 3 shows our results over the entire dataset. Comparing row 1 (GAMMA, as-is) and row 2 (GAMMA with Graph Attention replaced with a standard fully-connected linear layer), we clearly see that the performance numbers drop significantly, indicating the importance of Graph Convolution in the design of GAMMA. Similarly, comparing row 1 with row 3 (GAMMA with Self Attention removed), we again see a drop in performance for all three metrics. This highlights the significance of self-attention in the design of GAMMA.

We find that while replacing GAT with a linear layer, the dependency-agnostic representations are not as helpful for localizing bottlenecks as GAMMA loses the ability to factor-in the neighborhood interactions in its inference. Removing self attention also has an adverse effect on the performance of GAMMA as the model becomes myopic, limited by the filter-size of the convolution layers, and loses its ability to hold long-term patterns.

## 5 CONCLUSION

Online web applications are increasingly adopting the microservices architecture (MSA). While modular and flexible, MSA applications have numerous microservices that interact with each other in complex ways, making it difficult to identify and pinpoint performance bottlenecks. Further, since multiple bottlenecks can arise independently or dependently for an MSA application, it is crucial to accurately detect and localize *all* performance bottlenecks.

This work focuses on the key gap in this problem space—the ability to detect **multiple bottlenecks** efficiently for MSA applications, a realistic use-case that has been ignored by prior works. Our solution framework, GAMMA, learns complex interactions between microservices using graph neural networks and integrates this with a mixture of experts to enable multiple bottleneck localization. Evaluation results using the DeathStarBench Social Networking application highlight the superiority of GAMMA compared to several existing techniques. Further, our results show that GAMMA can be trained efficiently and performs well **across bottleneck types**, unlike existing techniques. Finally, GAMMA's model lends itself to **explainability**, making it practical for performance diagnosis.

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
