# OpenReview forum: "GAMMA: Graph Neural Network-Based Multi-Bottleneck Localization for Microservices Applications"
_ACM.org/TheWebConf/2024/Conference — TheWebConf24 Oral_

### Official Review · Reviewer_vR4f · 2023-11-13

**Novelty:** 4
**Technical Quality:** 4

**Review:**

\textbf{Summary}:
- They propose an explainable graph learning model that integrates MoE to detect multiple bottlenecks, as well as to localize the source of such bottlenecks. This is particularly important in microservice architectures since they are becoming the most popular and defacto choice for building large-scale web apps.
- Overall, I think it is an interesting paper that needs a better evaluation.

\textbf{Strength}
- There isn't much work on localizing multi-bottleneck anomaly detection and bottleneck localization for MSA, so it is interesting that this work is being performed.
- The paper is clear and easy to follow

\textbf{Weakness}
- Evaluation section:  the similarity in requests (e.g., post creation, reading timelines) makes the evaluation fragile. As noticed by the results in figure 3, the results for the 3 evaluated requests are very similar.
- Graph accuracy: The graphs in figure 3 do not seem correct, the F1 score is said to be 0.13 (namely 0.91 - 0.78) but the graph is clearly above that. The text seems to indicate big differences in the results presented in figures 3a, 3b and 3c, however, differences between GAMMA and e_diag appear consistent across the three request types.
- Explainability: Results in table 2 lack clarity.  In the three cases, CPU+MEM was marked as the source of saturation. One of the columns should have been Network in order to validate if in such a scenario the model keeps mentioning CPU+Mem as the likely source of interference. Moreover, Is this the resource utilization at the node level or at the microservice? How would you distinguish if the microservice or the host has saturation when having only 1 feature related to CPU utilization?
- I thought that the main of the study was to pinpoint the exact service that is producing the bottleneck so that remedy actions could be taken, however, the results seem to be solely about interference at the node level.

\textbf{Scope}
- The paper equates bottleneck detection to anomaly detection. I think it would have been best to call it bottleneck detection, which is a particular type of anomaly

**Questions:**

Please refer to the review.

**Reviewer Confidence:**

3: The reviewer is confident but not certain that the evaluation is correct

**Scope:**

4: The work is relevant to the Web and to the track, and is of broad interest to the community

---

### Official Review · Reviewer_3AyF · 2023-11-17

**Novelty:** 5
**Technical Quality:** 5

**Review:**

This paper proposes GAMMA, a graph NN model to detect and localize multi-bottleneck for microservices applications. The method first maps input metric features into multi-channel embeddings using dilated convolution. It then constructs the dependency graph and learns a graph representation using attention. The learnt representation is then fed to a detecter and a localizer for prediction. The paper also creates a dataset for evaluation, and make it open-source for future research in this community.
The strengths and weaknesses are listed as follows.

Strengths:
1. The paper is very well-writen. The motivation, problem description, techinical contribution and method design is very clear.
2. The problem of multi-bottleneck detection and localization is novel to me.
3. The effort in creating the dataset and making it open-source is impressive.

Weakness:
My main concern is about the evaluation part. First, the compared baselines seem to be a little bit outdated to me. Both Seer and $\epsilon$-diag are works in 2019. Second, a few details related to the dataset and the experiments are lacking. Please see questions for details.

**Questions:**

1. The compared baselines seem to be a little bit outdated to me. This paper has mentioned a few recent related works (after 2019). I am wondering the reason why not choosing them as the baseline models for comparison.
2. The paper claims that window size $\tau$ is trainable. Can you illustrate on how $\tau$ is trained?
3. The paper also mentions that $\alpha$ can be tuned in the loss fucntion. What is the value you are using in the experiment? Are there any insights on tuning $\alpha$?
4. Can you provide the details of the dataset you created ? (e.g. how many training and testing data points in the dataset? how is the train and test data split? what is the portion of window examples where there is zero/one/two... bottlenecks? etc.)
5. How to apply GAMMA to a real-world MSA application? I am interested in the general procedure (e.g. can GAMMA use existing public dataset for training or it also needs data from the real-world system? how is the generalization ability? where will GAMMA be deployed? etc.)

**Reviewer Confidence:**

2: The reviewer is willing to defend the evaluation, but it is likely that the reviewer did not understand parts of the paper

**Scope:**

4: The work is relevant to the Web and to the track, and is of broad interest to the community

---

### Official Review · Reviewer_qSoR · 2023-11-22

**Novelty:** 3
**Technical Quality:** 3

**Review:**

Paper Summary:
The paper proposes GAMMA, a new model for detecting and localizing performance bottlenecks in applications using microservices architecture. Microservices applications are prone to performance issues when multiple bottlenecks arise independently or dependently across services. Prior works focus on single bottlenecks and do not utilize distributed tracing well. GAMMA uses graph neural networks to learn complex interactions between microservices from distributed traces and metrics. It also uses a mixture of experts model to account for multiple bottlenecks. GAMMA is evaluated using a new dataset with 40 million request traces and multiple bottlenecks. Results show GAMMA significantly outperforms prior techniques like deep learning and statistical methods in both anomaly detection and bottleneck localization. GAMMA claims to achieve 46% higher F1 score than existing works that employ deep learning.

Strengths:
+	This paper studies an important question to detecting and localizing multiple performance bottlenecks in a holistic manner.
+	The proposed method GAMMA seems to have a remarkable performance gain.
+	The authors offer to provide a new dataset with 40 million request traces with respect to multiple bottlenecks.

Weaknesses:
+	Lack of technical novelty.
+	Important details of experiments are not described.
+	The explanation of the evaluation results is lack of reasoning.

Detailed comments:

It seems that GAMMA is a combination of a bunch of existing techniques such as dilated casual convolution, self-attention, and graph-attention network. More importantly, this combination seems work just fine to solve the problem, are there any challenges when applying these techniques? Or are there any other alternatives can be applied and why they are discarded?

Some important details of experiments are not described.
Above all, GAMMA is claimed to localize multiple bottlenecks, however, the authors do not mention in Sec. 4.2 about the distribution of the bottlenecks, i.e., within the 40 million request traces, what is the minimum and maximum number of the bottlenecks.
Although Figure 4 shows that GAMMA outperforms existing works, it is hard to prove the above claim without a more detailed comparison, e.g., use a benchmark with different number of bottlenecks.
Moreover, GAMMA and baselines are evaluated using three typical request patterns, called compose, home, and user. However, the authors do not provide details of these requests and why these three request patterns are representative.

The explanation of the evaluation results is lack of reasoning.
Throughout the Sec. 4.4, the authors keep saying that GAMMA performs consistently better than baselines in different scenarios without giving any reasoning or insights, making their claim less strong.
For example, in Figure 3, why epsilon_diag has an extreme poor F1 score and Recall throughout the three request types but GAMMA can perform consistently better? What is the key enabler technique here?
Another representative example is the explanation of Table 2. Why the F1 score is increasing when omitting the network feature? The direct conclusion is that the feature is harmful to the model performance (as mentioned by the authors). Then it seems that the author should not include this feature in some certain scenarios at all.

Minor issues:
-	Some notations are not explained in Equation 1 and 2.
-	The parentheses of the ReLU function in Equation 4 are missing.

**Questions:**

1. What is the key difference between single bottleneck identification and the multi-bottlenecks one? It seems that the single bottleneck identification model can be applied to the multi-bottlenecks scenario with minor changes (e.g., manipulating the threshold).

2. What are the key technical challenges when applying existing techniques (e.g., DCC, attention) to design GAMMA?

**Reviewer Confidence:**

3: The reviewer is confident but not certain that the evaluation is correct

**Scope:**

3: The work is somewhat relevant to the Web and to the track, and is of narrow interest to a sub-community

---

### Official Review · Reviewer_QAfZ · 2023-11-23

**Novelty:** 5
**Technical Quality:** 4

**Review:**

Microservices are becoming popular for large-scale online web applications. Performance management including anomaly detection and bottleneck localization is critical yet challenging. This paper targets this area and focuses on the multi-bottleneck scenarios in microservices applications. GAMMA is presented as a framework to learn the interactions between microservices using a GNN trained with distributed traces. The key is to understand call graphs with different system metrics and graph dependencies. The authors also generate a dataset containing a large amount of request traces, and compare GAMMA with other existing proposals with various evaluations. GAMMA is able to show its superior performance in detecting multiple bottlenecks.

The paper is well-written with a clear structure and detailed explanation of the proposed model and evaluation methods. GAMMA's approach to using graph neural networks for multi-bottleneck detection and localization in the microservices architecture is a novel contribution. Addressing multi-bottleneck detection is crucial for maintaining performance in complex web applications, making this research highly relevant and significant. The intention to open source the dataset can be beneficial to the research community.

Meanwhile, I would like to point out several points where improvements are needed. In the design, it is mentioned that GAMME employs a mixture of experts for bottleneck detection and localization. The description is a bit superficial. The paper would benefit from more details on the classifiers and localizers and the reasoning behind the specific structures.

Secondly, the complex nature of GAMMA might pose challenges in real-world implementation. The paper lacks details on the implementation of GAMMA. Also, the complexity could lead to higher computational resource requirements, impacting its feasibility in resource-constrained environments. It is unclear to me the computational overhead and how it affects different sizes of web applications.

While the model performs well in the presented scenarios, I would be interested in a thorough evaluation of scalability and adaptability to different application sizes and types.

Some discussions on potential overfitting when applied to less diverse or smaller datasets, would be beneficial.

**Questions:**

What do the anomaly detector and bottleneck localizer consist of? What is the rationale behind choosing them? I was hoping to get some details on this, as well as the implementation details. Otherwise, it would be difficult for others to reproduce and compare it with newly proposed solutions.

Can the authors elaborate on how GAMMA adapts to different scales of microservices architectures and varying operational conditions?

What measures were taken to prevent overfitting, given the model's complexity and the richness of the dataset used?

**Reviewer Confidence:**

2: The reviewer is willing to defend the evaluation, but it is likely that the reviewer did not understand parts of the paper

**Scope:**

4: The work is relevant to the Web and to the track, and is of broad interest to the community

---

### Official Review · Reviewer_3Y4t · 2023-11-25

**Novelty:** 5
**Technical Quality:** 6

**Review:**

Multi-bottleneck localization for microservice applications is important and challenging work in the domain of microservices research. GAMMA clearly indicates the problem it targets to solve, and the overall design and evaluation are persuasive to convince that GAMMA outperforms the baselines.

Pros

-End-to-end joint training framework that learns complex interactions between microservices using an attention-based graph convolution. It makes decisions whether the application itself is in an anomalous situation, followed by bottleneck localizer that detects all the multiple bottlenecks that are causing the anomalies.

-GAMMA incorporates many recent machine learning techniques in different stages of model, specifically designed to work along with microservices application characteristics. It utilizes multi-source temporal embeddings to capture temporal pattern and encapsulate spatial pattern. They are fed to Dilated Causal Convolution for efficient capturing of relation within and between features over time. Later, Self-attention and Graph-attention is applied for graph representation learning. Based on those representations, each expert specialized for each different microservice decide rather the microservice is the bottleneck or not.

-Generated a dataset consisting of around 40 million request traces, and open-sourcing the dataset. Any form of rich open-sourced anomaly dataset is helpful to the microservice research community.

Cons

-FIRM does detect multiple anomalies with SVM and provides evaluations on it. Although GAMMA outperforms FIRM in detecting multiple anomalies, it seems difficult to claim GAMMA is the first and only work that solves the problem of finding multiple bottlenecks. However, such a claim is mentioned several times throughout the paper.

-4.4.5 Explainability of GAMMA section does not provide useful information about explainability of GAMMA. It is obvious that ML model would suffer performance degradation when features that are directly related to the predictions are excluded. The generic term of explainable AI is used in the case where algorithms provide the reasoning behind a given model’s decision-making. Can GAMMA provides certain form of processed data or confident rates behind each decision they make?

**Questions:**

-How do you define or set the ground truth for the dataset used for training? (The ground truth for the bottlenecks and anomalies from the trace data.) Are there cases that many dependent microservices are suffering very slight bottlenecks in terms of the load, so there are no particular bottlenecks to be detected yet performance anomalies take place? How does GAMMA act in such cases?

-In what specific cases do baselines (FIRM, Seer) fail to predict while GAMMA successfully predict the multi bottlenecks? Can you categorize characteristics of those specific cases where baselines fail and GAMMA outperforms?

-The Social Network consists of 28 microservices which are relatively small to modern microservice applications that are reported to extend to hundreds (e.g., Netflix, Amazon). While GAMMA input feature vectors increase linearly to the number of microservices in the application. To what number of microservices would GAMMA be scalable to predict? Is it possible that GAMMA performs worse than baselines (FIRM, Seer) when the number of microservices increase?

**Ethics Review Description:**

No issues.

**Reviewer Confidence:**

4: The reviewer is certain that the evaluation is correct and very familiar with the relevant literature

**Scope:**

4: The work is relevant to the Web and to the track, and is of broad interest to the community

---

### Decision · Program_Chairs · 2024-01-22

**Decision:**

Accept (Oral)

**Comment:**

The paper addresses a GNN-based framework to locate bottlenecks in the microservices environment. The paper evaluates the proposed framework, GAMMA, with multiple baselines and demonstrates the advantages.

 The reviewers are mostly positive about the paper, appreciating the novelty in its problem statement and approaches, end-to-end coverage of the proposed framework, extensive evaluation and production of large-scale datasets. In the meantime, the reviewers also pointed out several rooms for improvement, such as justification of the baseline in the experiments, explainability of the proposed framework, complexity of the system in consideration of real-world deployment. I also greatly appreciate the authors' effort in thoroughly communicating with the reviewers.

 Overall, given the ratings, review comments, and discussion, I believe that this paper's merits certainly outweigh its concerns. My AC recommendation is set accordingly.